# County medical community, medical insurance package payment, and hierarchical diagnosis and treatment—Empirical analysis of the impact of the pilot project of compact county medical communities in Sichuan Province

**Shaoqun Ding[1¤], Yuxuan Zhou[2¤]***

1 Research Center for Aging and Social Security, Southwestern University of Finance and Economics, Chengdu, Sichuan, China, 2 School of Public Administration, Southwestern University of Finance and Economics, Chengdu, Sichuan, China

¤ Current address: Liulin Campus, Southwestern University of Finance and Economics, Wenjiang District, Chengdu, Sichuan, China

* 15556932867@163.com

**Data Availability Statement:** All original data relevant to this paper can be accessed through the following links for free: http://wsjkw.sc.gov.cn/

## Abstract

Hierarchical diagnosis and treatment (HDT) is an important exploration direction to alleviate the rising pressure of health expenses and medical insurance fund expenditure in China, and to maintain and protect the public health in this country. In recent years, the construction of compact county medical communities (CCMC) has become the primary approach for implementing the HDT. Utilizing the quasi-natural experiment of the pilot project of CCMC in Sichuan Province in 2019, coupled with county-level data extracted from the ' Sichuan Provincial Health Statistics Yearbook ' spanning the years 2008 to 2021, this research evaluates the effect of the pilot project of CCMC on promoting HDT under the medical insurance package payment model. The results show that the pilot project of CCMC has significantly increased the number of consultations per capita of medical and health institutions in pilot counties by 0.434 times, of which the number of consultations per capita of primary medical institutions has increased by 0.340 times; the number of hospitalizations per capita in public hospitals and primary medical institutions in pilot counties increased significantly, and the surgery rate of inpatients in public hospitals increased by 5% compared to before the pilot. There was no significant impact on the allocation of medical facilities and human resources in the pilot counties. Therefore, the construction of CCMC under the medical insurance package payment mode has promoted the realization of the county-level HDT. These findings provide valuable insights for healthcare policy, especially in developing and implementing effective strategies for HDT in county-level medical institutions.

scwsjkw/njgb/tygl.shtml and https://www.wind.com.cn/portal/zh/EDB/index.html; the data that have been organized are included in the paper and its Supporting information files "S4_File. Data sample. (xlsx)

**Funding:** National Social Science Fund Major Project ' Research on Multi-level Social Security System Innovation and Policy Synergy Based on System Concept ' (23ZDA099); the basic scientific research business fee of the central university ' Research on the effect of urban and rural residents ' medical insurance outpatient co-ordination to promote hierarchical diagnosis and treatment ' graduate research project (JBK2307025).

**Competing interests:** The authors have declared that no competing interests exist.

# 1.Introduction

China's reform of HDT (HDT refers to the grading of diseases according to their severity and ease of treatment, with medical institutions at different levels undertaking the treatment of different diseases and gradually realizing the process of medical treatment from general practice to specialization.) primarily relies on the medical consortium model to achieve its goals [1]. The concept of a medical consortium is rooted in the global practice of Integrated Healthcare, which involves the collaboration of multiple medical institutions. By leveraging the complementary strengths and the judicious allocation of resources among these medical service providers, a medical consortium offers patients a seamless continuum of healthcare services, enhancing service quality while reducing costs. This, in turn, seeks to optimize the utilization of medical resources [2].

In 2015, The General Office of the State Council in China released the "Guiding Opinions on Promoting the Construction of a HDT System," which advocated the exploration of medical consortium formation and the gradual development of a regional collaborative service model with clearly defined roles, rights, and benefits. Since the inception of the medical consortium construction in 2015, various pilot areas have ventured into diverse modes of exploration. Depending on the closeness and scope of implementation of the medical association, these include early trusted-type county medical communities, loose urban medical communities, and loose county medical communities [3]. Notable examples of medical community models include those implemented in Tianchang, Anhui, Funan, Anhui, and Youxi, Fujian.

In April 2017, The General Office of the State Council issued the "Guiding Opinions on Promoting the Construction and Development of Medical Consortia," which introduced the concept of four distinct medical consortia, namely, city medical groups, county medical consortia, inter-regional specialty alliances, and telemedicine collaboration networks, each with its specific functional roles. The focus was on the exploration of the CCMC which integrated management at the county and township levels, with county-level public hospitals taking the lead, township health centers serving as the central hubs, and village clinics forming the foundational basis. This approach aimed to establish a county-level medical service system characterized by seamless collaboration and coordination across all three levels, from counties down to villages. Consequently, in 2017, pilot regions initiated their efforts to explore the establishment of tightly-knit medical communities.

In June 2017, The General Office of the State Council released the "Guiding Opinions on Further Deepening the Reform of Basic Medical Insurance Payment Methods." This document proposed the exploration of implementing comprehensive medical insurance prepayment for vertically aligned cooperative medical unions, including the CCMC, and other collaborative mechanisms. In May 2019, the National Health Commission and the State Administration of Traditional Chinese Medicine issued a notice on promoting the construction of CCMC that urged the enhancement of various payment methods, such as comprehensive medical insurance payments. It also encouraged the exploration of total head budget management for medical insurance, along with the establishment of mechanisms to retain surplus funds and share reasonable expenditure.

Subsequently, in August 2020, the National Health Commission and other relevant departments issued a notice introducing the Evaluation Criteria and Monitoring Index System for the Construction of CCMC (Trial). This notice explicitly called for the development of payment policies tailored to the specific characteristics of CCMC and the exploration of comprehensive payment mechanisms facilitated by the medical insurance fund for CCMC. The reform of the "Medical Insurance Package Payment," a multifaceted and intricate medical

insurance payment approach within the overarching comprehensive medical insurance system, plays a pivotal role in providing substantial support for the development of CCMC.

Following the issuance of the Notice on Promoting the Construction of CCMC in 2019, the National Health Commission initiated a nationwide pilot project for the establishment of such communities. It identified 754 counties in Shanxi and Zhejiang provinces, along with 567 counties in other provinces as pilot counties. In September 2019, the Sichuan Provincial Health Commission, in collaboration with other relevant departments, released the "Implementation Plan for the Construction of CCMC in Sichuan Province" (referred to as the "Implementation Plan" hereinafter). This plan drew on the earlier nationwide pilot experiences in policy formulation, refining the roles and authority of each system participant.

Differing from the province-wide implementation of the compact county health community pilot in Shanxi and Zhejiang provinces, Sichuan Province, which had not previously conducted such a pilot, selected specific counties to implement the pilot project across the entire province leaving others without any pilot project intervention, providing a valuable experimental group and control group for this study.

As a unified medical organization alliance, the CCMC is anticipated to optimize the integration of medical resources, achieve efficient resource allocation, enhance grassroots service capabilities, and promote the realization of a HDT system [4,5]. Therefore, the key questions of interest in this paper pertain to the effectiveness of the CCMC pilot in Sichuan Province in 2019.

Firstly, can the CCMC, dedicated to enhancing primary care services, effectively guide patients to seek medical care at the county level? Secondly, the establishment of CCMC has transformed the competitive relationships between medical institutions at all levels into cooperative ones. Will this affect the diagnosis and treatment practices of these institutions at various levels? Finally, can the creation of CCMC genuinely improve the allocation of medical resources, including health personnel and medical facilities?

These are crucial questions addressed in this paper. Accordingly, relying on data from the Sichuan Provincial Health Statistical Yearbook spanning from 2008 to 2021, the paper employs the PSM-DID (Propensity Score Matching and Difference-in-Differences) method to assess the effectiveness of the pilot counties in the implementation of CCMC in Sichuan Province.

The other sections of this paper are structured as follows:

Background and Literature Review: This section delves into the historical evolution of HDT both domestically and internationally. It also provides an overview of the reforms in medical insurance payment methods, the restructuring of medical consortia and medical communities, and the assessments and appraisals by scholars, both national and international, on the establishment of an integrated medical system and the role of medical unions and medical communities.

Theoretical Mechanisms: This segment outlines the optimal decisions made by each participant in the construction of CCMC. It explores how these decisions align with the incentive compatibility mechanism of "medical insurance package payment" and the internal reforms of CCMC.

Empirical Analysis: In this part, the paper details the sources of data used, the selection of variables, and the choice of models for analysis. It further presents the empirical findings on various topics of interest within the paper and concludes with policy recommendations and conclusions derived from the empirical results.

The structure of this paper provides a comprehensive examination of the development and implications of HDT, with a focus on the implementation of CCMC and their effects within the context of evolving medical insurance payment systems.

## 2.Literature review

The structure of this paper provides a comprehensive examination of the development and implications of HDT, with a focus on the implementation of CCMC and their effects within the context of evolving medical insurance payment systems.

The HDT system entail ensuring that all patients receive the appropriate care at the right time and in the right healthcare setting. One of its key benefits lies in providing patients with continuous, high-quality care at a relatively low cost [6]. In China, healthcare providers are categorized into two primary tiers: primary medical institutions and hospitals. The former, typically offering general medicine services, primarily handle the diagnosis and treatment of common and frequently encountered health conditions, while the latter, housing specialized departments and more specialized physicians, focus on diagnosing and treating complex and severe diseases.

From the provider's perspective, the HDT system is orchestrated by the health department. This involves the implementation of policies such as enlisting family doctors as gatekeepers, the establishment of medical unions and medical communities, and the reform of the payment mechanisms within basic medical insurance.

On the demand side, the basic medical insurance system should improve differentiated payment structures, prompting urban and rural residents to proactively embrace HDT [7,8]. Encouraging the public to engage in graded diagnosis and treatment through the implementation of varied reimbursement policies in medical facilities at different levels has become a widely adopted strategy across most regions of the country [9].

However, it is worth noting that this approach often relies on economic incentives and tends to overlook both the psychological aspects of healthcare decision-making among individuals and the phenomenon of tertiary hospitals disproportionately attracting patients. Because people tend to skip those primary level institutions and go straight to higher level hospitals, for ailments or diseases sometimes of easy diagnosis and treatment or management, those primary care facilities end up having very small number of patients, hence not being cost-effective. The profit-driven motives of medical institutions and the limited capacity of primary healthcare facilities pose significant challenges in realizing the vision of comprehensive HDT [10].

Research conducted by Shen and Zhang (2016) in the Pearl River Delta region of China reveals that residents' inclination to seek medical care at the primary level is notably low [9]. This is primarily attributed to the insufficient capacity and resources of primary medical facilities, which, in turn, erode trust in the concept of "primary first diagnosis." Further investigations have unveiled several underlying issues. Policy-driven primary medical institutions prioritize public health over medical care, and this is coupled with a lack of robust performance incentives, an incomplete compensation structure, antiquated salary and management systems within these primary healthcare facilities. These factors collectively contribute to the diminished motivation among primary healthcare staff [11–13]. Therefore, the establishment of a HDT system faces significant hurdles, with the root cause residing in the inadequacies of medical care, rather than medical insurance [14]. But also see the aspect of people wishing to go above primary levels.

In the 1990s, the United States witnessed the emergence of the concept and practice of "integrated healthcare" on the supply side of the medical field. This approach involved the consolidation of various healthcare service providers into a coordinated network, with the aim of delivering comprehensive, end-to-end healthcare services to patients, encompassing prevention, diagnosis, treatment, and rehabilitation [15]. Several studies have demonstrated the effectiveness of Accountable Care Organizations (ACOs), a form of American medical consortium,

in proactively controlling medical costs without compromising the quality of patient care [16,17]. Medicare in the United States has augmented incentives to control costs by innovating the payment model for medical consortia, thereby encouraging cost-effective healthcare delivery [18]. Experience shows that: a healthy institutional collaboration will help overcome the development loopholes in Asia, promote sustainable public health quality [19]. A sound and effective policy enforcement by enhancing resource capacity, quality of institutional practices, will promote Sustainable Development Goals [20].

In recent years, the medical commonwealth model has emerged as a prominent representative of the reform strategies adopted by public hospitals. It is widely regarded by a majority of scholars as "the strategic choice for China's healthcare reform" and a pivotal measure in the ongoing and comprehensive supply-side structural transformation of China's healthcare system [21,22]. However, in the early stages of its implementation in China, the member institutions within the medical commonwealth were integrated mainly in form, resulting in a dispersion of actual power, responsibility, and benefits. The lead hospitals often lacked effective incentive mechanisms, leading to reduced operational efficiency [23].

From a localized perspective, the CCMC stands out as the primary model for medical commonwealth construction at present. Its feasibility and practicality make it a significant breakthrough in the pursuit of an efficient and high-quality medical and healthcare service system. It bears practical significance in addressing the fragmentation within the healthcare service system and enhancing the overall efficiency of county-level medical and healthcare resources. As a result, the current implementation of the medical community model has evolved over time. It has transitioned from a diverse array of pilot projects at the outset to exhibit the characteristics of a more cohesive and compact medical community model [24].

The medical community introduces a multi-level principal-agent relationship involving "patient—medical insurance—hospital—doctor." However, the establishment of a tightly-knit medical community addresses the challenges posed by this complex relationship through a process of "integration." This integration aligns the interests of all parties within the medical system, transforming them into a "community of interests" that collectively serves the overarching objective of the entire system: maintaining the health of the entire population at the lowest possible cost [25,26].

Building upon this foundation, policy entrepreneurs in the early stages of the "bottom-up" development of the medical community collaborate with internal and external mechanisms once the "top-down" policy is later implemented. By reforming the medical insurance payment system, personnel performance evaluation and establishment mechanisms, family doctor sign-up procedures, and information sharing protocols, a new incentive compatibility mechanism is constructed to align the interests of government, hospitals, doctors, and patients. As a result, this restructuring leads to Pareto improvements (Pareto Improvement refers to a situation where the welfare of at least one individual is improved without worsening the welfare of any other individual [27]). and enhanced healthcare outcomes, ultimately facilitating the establishment of an effective HDT system [2,28].

The existing literature primarily focuses on assessing the operational outcomes of local medical communities, with evaluations centered on the hypothesis that medical communities promote HDT and optimize the allocation of medical resources, ultimately reducing the economic burden on patients and the medical insurance fund.

Firstly, the establishment of medical communities serves to enhance the allocation of healthcare resources by fostering two-way referral channels (Typically, primary care facilities refer patients with complex conditions or in need of specialized treatment to higher-level hospitals; after the specialized treatment, higher-level hospitals refer patients back to primary care facilities for follow-up rehabilitation or long-term treatment), strengthening the capacity of

primary medical institutions, instilling trust in these institutions among patients, and establishing a positive feedback loop [29]. When it comes to hospitals, studies indicate that the implementation of medical communities in certain cities has enhanced the medical service capabilities of member institutions, bolstered overall competitiveness, improved medical quality, and facilitated two-way referral services [30]. It has been verified that the construction of a medical community in a sub-provincial city contributes significantly to HDT, with secondary hospitals benefiting the most [31].

In the case of primary medical institutions, the example of Zhongxian in Chongqing, which introduced the concept of a "vertical and horizontal union" medical community, offers a notable illustration. This reform led to increased business capacity of primary medical institutions by 25%, a 38% rise in the number of licensed physicians, an 84% increase in the average salary of medical staff, a 3.3 percentage point rise in the county-level consultation rate, and an 8.5 percentage point increase in the first-visit rate [32]. Empirical analyses conducted by Wang and Sun (2021) on the HDT reform implemented in Shanghai in 2015, featuring family doctor contracts and unstructured medical unions, revealed a significant impact of the diagnosis and treatment capabilities and the treatment environment of primary medical institutions on patient satisfaction, consequently enhancing patient loyalty [33].

The CCMC in Tianchang has achieved a comprehensive medical service governance mechanism that involves "hospitals, patients, doctors, and medical insurance." It has also laid the groundwork for a rural healthcare system centered on residents' health, characterized by the ability to receive initial diagnoses within 15 minutes and access specialized care within 50 kilometers [34].

A five-year reform initiative in the construction of a CCMC in Nagqu, Tibet, has significantly increased the availability of high-quality medical resources, creating a three-tier county-level medical service system that connects counties and villages. This has led to a 53% reduction in referrals outside the county, essentially realizing the goals of keeping minor ailments at the township level and managing moderate diseases at the county level [35].

In terms of alleviating the financial burden on patients and optimizing the medical insurance fund, a case analysis of Yunxian in Yunnan conducted by Zhu and Duan (2021) has yielded impressive outcomes [36]. The Yunxian medical community reform not only established a HDT pattern but also significantly reduced the average medical treatment costs for residents while enhancing the efficiency of the medical insurance fund. In 2020, the balance of the medical insurance fund in Yun reached nearly 20 million yuan, a substantial improvement compared to the overall city budget, which was incurring a deficit of approximately ¥1 million.

Furthermore, Liao et al (2023) leveraged the operating data from 12 CCMC pilot counties in Sanming, which implemented a total medical insurance fund underwriting combined with C-DRG (Case-Based Payment System) collection and payment [37]. Their extensive evaluation of the impact of medical insurance package payment reform on the operation and management of CCMC demonstrated that this reform, the CCMC could alleviate the financial burden on patients, increase hospital staff salaries, and bolster the balance of the medical insurance fund.

However, it's important to acknowledge that there remain several challenges in the local implementation of medical communities. Firstly, the top-level design of policies often lacks clear objectives and comprehensive plans, with insufficient systematic support from related policies. This frequently results in a situation where policies are symbolically implemented but lack substantial impact [38,39].

Secondly, the process of organizational integration sometimes fails in terms of resource allocation. Weaker grassroots institutions may find their resources further concentrated in

more robust superior institutions, exacerbating the imbalance in healthcare provision, with the strong becoming stronger and the weak becoming weaker [10].

Additionally, the lack of effective incentive-compatible design hampers the motivation of partners to participate in these alliances and share resources. This can result in a distortion where the nominal collaboration actually fosters competition [40,41]. While some studies have shown that medical union reforms can enhance the efficiency of community medical and healthcare services to some extent, the implementation of these policies often increases the number of patients seeking care at community facilities without significant improvements in service quality or patient satisfaction [42].

At the present stage, system design, benefit distribution, information sharing, common resource management, and supervision methods all exhibit deficiencies. These shortcomings lead to high transaction costs in aspects like incentive management, coordination, information sharing, decision-making, and oversight. Addressing these challenges is crucial for the effective and efficient functioning of medical communities [43].

Based on a field investigation of 11 pilot counties of CCMC in Zhejiang, Xu and Yu (2020) identified the reform of the medical insurance payment system as a critical factor influencing the success or failure of CCMC [44]. In Zhejiang, medical insurance payments continue to be primarily based on a post-payment system controlled by total expenditure. Project-based payments still dominate, with incremental budgeting predominantly determining the total budget base. The incentive mechanisms related to sharing overspending and retaining surpluses need improvement, significantly constraining the realization of the policy objectives of the medical community.

Wu and Li (2023) found that the implementation of medical insurance package payment with CCMC in Fujian encouraged patients to seek treatment within the county but did not significantly promote patient triage to grassroots-level healthcare facilities [45]. In practice, the impact of medical insurance balances on motivating collective action is relatively weak, and the effectiveness of supporting policy tools such as medical insurance payment rates and salary structures in aligning individual behavior with collective goals is limited [46].

The total prepayment system has played a positive role in controlling medical expenses within county medical communities and preventing excessive healthcare behaviors. However, due to the absence of internal incentive and restraint mechanisms, member institutions may still decline patients, encourage patient transfers to other hospitals, prefer non-medical insurance patients, and avoid the use of expensive medications and high-value consumables, actions that are inconsistent with their intended roles and functions [47].

Most of the existing literature is focused on describing the policies and the current state of medical union and medical community construction. Early studies primarily involved evaluating the effects through investigations of individual urban medical community pilot programs, with limited use of empirical testing.

The theoretical significance of this paper lies in its effort to rationalize the incentive compatibility mechanism among participants in various systems created by the reform of package payments for medical insurance in Sichuan Province and the internal reform of medical communities. It contributes to enriching the theory related to HDT in China.

The practical significance of this paper is that it employs a rigorous approach, combining sample selection with the differential difference (PSM-DID) method and reliable data to investigate the impact of the CCMC in 37 counties of Sichuan Province on patients' choices for medical treatment, hospital diagnosis and treatment behaviors, and medical resource allocation. This research offers valuable insights on how to further enhance the construction of CCMC and promote the HDT system in China.

## 3.Theoretical mechanism analysis and research hypothesis

The concept of "incentive compatibility" was initially introduced by Hurwicz (1973) [48]. In economic activities, when there is an institutional arrangement that enables individuals to align their pursuit of self-interest with the objective of maximizing collective value, it is referred to as "incentive compatibility." This theory was first applied by Mauris (1997) in England to address principal-agent problems in contract design [49].

The incentive compatibility framework for CCMC in Sichuan Province, aimed at achieving graded diagnosis and treatment under the medical insurance package payment system, is depicted in Fig 1. This framework comprises four institutional actors: government departments, the CCMC, doctors, and patients, and three principal-agent relationships, which are patient-government departments, government-CCMC, and CCMC-doctor relationships.

The Implementation Plan strives to effectively align the interests of principals and agents at various levels of the principal-agent chain through a range of policy measures and system optimizations. The objective is to motivate each institutional actor to pursue their self-interest while simultaneously advancing the collective goal of maximizing overall value.

The construction of CCMC in Sichuan Province involves four institutional subjects and three levels of principal-agent relationships. In these relationships, there is a significant information asymmetry and misalignment of objective functions between principals and agents within the traditional non-medical community model. Additionally, the government departments, the CCMC, and the medical practitioners exist at different levels or within different departments, making the internal and external relationships of the medical community complex.

The establishment of CCMC in Sichuan Province relies on various factors, including the reform of the medical insurance payment method, enhancements to the employment mechanism, adjustments to the salary incentive mechanism, improvements in the family doctor signing system, modifications to medical service pricing, the comprehensive allocation of resources, the creation of smooth two-way referral pathways, the promotion of the overall integration of medical and preventive services, the enhancement of information systems, and the implementation of financial investments and supervisory responsibilities. These efforts collectively contribute to the creation of an incentive compatibility mechanism for HDT.

The basic medical insurance system is the largest financial contributor to medical institutions. A well-designed medical insurance system can effectively influence and regulate patients' healthcare-seeking behaviors [50], as well as motivate doctors at various levels of healthcare facilities [51], including grassroots hospitals, to actively participate. This is conducive to achieving the reform objectives of graded diagnosis and treatment.

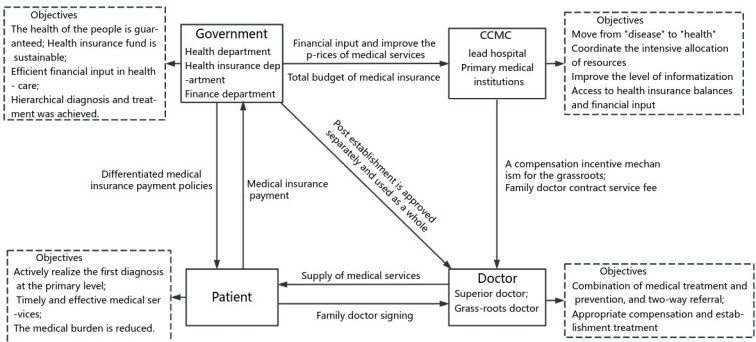

**Fig 1. Incentive compatibility framework of CCMC to realize HDT under "package payment" of Sichuan medical insurance.**

In the realm of medical insurance system design, payment system reform is pivotal to the success or failure of CCMC [44]. In Sichuan Province's "implementation plan," key proposals include: enhancing the total budget management of the medical insurance fund; establishing the mechanism of "surplus retention", and "reasonable overspending sharing." adjusting the total control indicators to be more favorable to grassroots medical and healthcare institutions; allowing CCMC that have implemented financial, information, and settlement integration to implement a "unified total amount" for medical insurance and adjust the total amount index within the CCMC; implementing a comprehensive "multi-compound medical insurance payment method based on disease type." These measures collectively address the core aspects of the medical insurance system and its payment mechanisms, contributing to the promotion of graded diagnosis and treatment reform.

In terms of asset management, the CCMC is authorized to coordinate the deployment of medical equipment and facilities in accordance with relevant regulations on asset disposal. Concerning informatization, the policy promotes the integration of information systems across all member units of the medical community, facilitating data sharing and medical insurance settlement integration, which creates the conditions for the realization of total medical insurance advancement. These efforts aim to enhance the digital operations of CCMC, such as resource allocation, business operations, and efficiency monitoring. They also encourage the continuous recording of electronic medical records and electronic health records.

The medical insurance package payment policy, combined with the construction of CCMC, encourages a shift from focusing solely on "disease" to a broader focus on "health." The delegation of asset management authority and the advancement of informatization levels incentivize the CCMC to efficiently share and allocate medical resources between primary and higher-level medical institutions. This leads to cost reduction, improved medical service effectiveness, and the attainment of medical insurance balance [52].

Doctors play a central role in providing medical services and are key to achieving the goals of all stakeholders in CCMC. The "Implementation Plan" grants the medical community full autonomy in internal management, giving them the authority to manage staffing and salary distribution. It encourages the establishment of a flexible employment mechanism that allows for adjustments in staffing levels, facilitates job rotation, and promotes two-way personnel flow within CCMC. Additionally, the plan promotes the adoption of a contract service fee for family doctors, shifting the focus of the CCMC from treating pre-existing diseases to giving equal attention to both pre-existing and non-existing diseases. It also suggests a salary distribution mechanism that prioritizes clinical front-line and grassroots medical staff, with reasonable wage gaps.

These measures, such as the flexible employment mechanism, family doctor contract service fees, and the distribution of salaries favoring grassroots medical staff, encourage doctors from higher-level hospitals to work at grassroots levels and motivate grassroots doctors. This promotes the realization of "two-way referral" and the integration of medical and preventive services within the CCMC [2].

Under the construction of CCMC, there has been an improvement in the medical service capacity of grassroots medical institutions. The "Implementation Plan" further recommends enhancing differentiated medical insurance payment policies to guide insured individuals to prioritize their initial diagnosis at the grassroots level. For referred hospitalized patients who meet the criteria, continuous calculations of the starting payment line are suggested. Moreover, the key component of the CCMC's construction, the family doctor contract system [2], along with differentiated medical insurance payment policies, encourages insured patients to actively seek initial diagnoses at the grassroots level [53,54].

Regarding financial input, the plan states that subsidy funds for member units of CCMC should be allocated according to the CCMC's construction and development. Additionally, it suggests that basic public health service funds should be managed and used by CCMC as a whole, following the total budget of the permanent population served by the community. This approach ensures a stable source of funding for CCMC's development and construction while promoting the integration of medical service provision and public health service provision.

Local government departments also bear the responsibility of improving the pricing of medical services, supervising the quality of healthcare, and enhancing the assessment and evaluation system for the construction of CCMC. These policies collectively provide strong support for the construction of CCMC, ensuring the people's health is safeguarded, the sustainability of the medical insurance fund, efficient financial subsidies, and effective public health investment. Ultimately, these measures aim to achieve the goal of HDT.

The "Implementation Plan" has established specific work goals for the construction of CCMC, with a focus on increasing consultation rates and enhancing the role of grassroots medical institutions in providing healthcare. These goals include striving for a 90% consultation rate at the county level and increasing the proportion of outpatient, emergency, and inpatient visits at grassroots medical institutions. To evaluate the effectiveness of these goals and the incentive compatibility mechanism framework, several hypotheses are proposed:

H1: The county-level consultation rate of patients in the pilot counties of CCMC increased, and more people proactively sought initial diagnoses at grassroots medical institutions.

H2: The medical service capacity of the pilot counties of CCMC has improved, with medical institutions at all levels effectively fulfilling their functional roles. This would result in the medical treatment pattern of "major diseases go to the hospital, minor diseases go to the grassroots medical institutions."

H3: Medical institutions at all levels within the pilot counties of CCMC have successfully optimized and allocated resources more reasonably.

These hypotheses will be tested to determine the impact of the CCMC's construction on the healthcare system and the achievement of HDT.

## 4.Data, variables and model

### 4.1. Data source

The data sources used in this paper are essential for conducting empirical research on the effects of pilot project of CCMC in Sichuan Province. The main data sources are as follows:

Sichuan Provincial Health Statistical Yearbook: This publication is a critical source for collecting data related to healthcare services, infrastructure, and various health-related indicators. It provides a comprehensive overview of the healthcare system in Sichuan Province and contains historical data from 2008 to 2021.

Wind Database: The Wind database is a financial and economic database that contains a wide range of financial and economic data for Chinese provinces and regions. In this paper, it was used to obtain various control variables that could potentially impact the performance of the pilot compact county medical communities.

"Implementation Plan for the Construction of Pilot Medical Community in Sichuan Province": This official document is used to identify which counties in Sichuan Province were designated as pilot areas for the construction of compact county medical communities. Table 1 serves as a reference for the selection of pilot counties, and Table 2 serves as a reference for the selection of non-pilot counties.

**Table 1. List of pilot counties for the construction of CCMC in Sichuan Province.**

| City | Number | List of counties | City | Number | List of counties |
|---|---|---|---|---|---|
| Chengdu | 5 | Xindu, Qingbaijiang, Qionglai, Xinjin and Pujiang | Yibin | 3 | Xingwen, Jiang 'an, Junlian |
| Zigong | 2 | Gongjing, Ziliujing | Guang'an | 2 | Guang 'an, Huaying |
| Panzhihua | 2 | Miyi, Yanbian | Dazhou | 1 | Tongchuan |
| Luzhou | 2 | Lu, Hejiang | Bazhong | 1 | Pingchang |
| Deyang | 4 | Mianzhu, Guanghan, Zhongjiang and Luojiang | Ya'an | 1 | Shimian |
| Mianyang | 1 | Pingwu | Meishan | 1 | Qingshen |
| Guangyuan | 1 | Chaotian | Ziyang | 1 | Anyue |
| Suining | 1 | Anju | Aba | 2 | Wenchuan,Jiuzhaigou |
| Neijiang | 1 | Longchang | Garzi | 2 | Ganzi, Seda |
| Leshan | 1 | Muchuan | Liangsha | 2 | Dechang and Xichang |
| Nanchong | 1 | Shunqing | Total | 37 | |

The data from these sources are crucial for conducting a comprehensive and data-driven analysis of the impact of the pilot compact county medical communities. Researchers can use this data to examine various variables and indicators that are relevant to the study's objectives and hypotheses.

## 4.2.Variable settings

The paper's research involved a range of explanatory variables, dependent variables, and control variables. These variables are essential for analyzing the impact of CCMC in Sichuan Province. It is important to note that data availability and consistency are crucial for conducting robust empirical research. Table 3 shows some key points regarding variable settings in the paper:

**Table 2. List of non-pilot counties for the construction of CCMC in Sichuan Province.**

| City | Number | List of counties | City | Number | List of counties |
|---|---|---|---|---|---|
| Chengdu | 15 | Jinjiang Qingyang Jinniu Wuhou Chenghua Longquanyi Wenjiang Shuangliu Pixian Jintang Dayi Dujiangyan Pengzhou Chongzhou Jianyang | Yibin | 7 | Cuiping Nanxi Xuzhou Changning Gaoxian Gongxian Pingshan |
| Zigong | 4 | Da'an Yantan Rongxian Fushun | Guang'an | 4 | Qianfeng Yuechi Wusheng Linshui |
| Panzhihua | 3 | Dongqu Xiqu Renhe | Dazhou | 6 | Dachuan Xuanhan Kaijiang Dazhu Quxian Wanyuan |
| Luzhou | 5 | Jiangyang Naxi Longmatan Xuyong Gulin | Bazhong | 4 | Bazhou Enyang Tongjiang Nanjiang |
| Deyang | 2 | Jingyang Shifang | Ya'an | 7 | Yucheng Mengshan Xingjing Hanyuan Tianquan Lushan Baoshan |
| Mianyang | 8 | Fucheng Youxian Anzhou Santai Yanting Zitong Beichuan Jiangyou | Meishan | 5 | Dongpo Pengshan Renshou Hongya Danleng |
| Guangyuan | 6 | Lizhou Zhaohua Wangcang Qingchuan Jiange Cangxi | Ziyang | 2 | Yanjiang Lezhi |
| Suining | 4 | Chuanshan Pengxi Daying Shehong | Aba | 11 | Ma'er'kang Lixian Maoxian Songpan Jinchuan Xiaojin Heishui Rangtang A'ba Ruo'er'gai Hong yuan |
| Neijiang | 4 | Shizhong Dongxing Weiyuan Zizhong | Garzi | 16 | Kangding Luding Danba Jiulong Yajiang Daofu Luhuo Xinlong Dege Baiyu Shiqu Litang Batang Xiangcheng Daocheng Derong |
| Leshan | 10 | Shizhong Shawan Wutongqiao Jinkouhe Jianwei Jingyan Jiajiang Ebian Mabian Emeishan | Liangsha | 15 | Muli Yanyuan Huili Huidong Ningnan Puge Butuo Jinyang Zhaojue Xide Mianning Yuexi Ganluo Meigu Leibo |
| Nanchong | 8 | Gaoping Jialing Nanbu Yingshan Peng'an Yilong Xichong Langzhong | Total | 146 | |

**Table 3. Variables and index systems of promoting HDT by county medical communities.**

| Variables | Index | | Details |
|---|---|---|---|
| Explanatory Variables | Medical community pilot system or not(ifygt) | | 0–1 variables, ifygt = 1, indicates the pilot counties of CCMC; ifygt = 0 indicates non-pilot counties. |
| Dependent Variables | Residents' choice of medical treatment | The number of consultations per capita (*dtp*) | The number of consultations include outpatient consultations, emergencies consultations, office consultations, individual health examinations, and health counseling and guidance consultations. The number of consultations per capita in medical and health institutions indicates the frequency of residents going to medical and health institutions for medical service; The number of consultations per capita to the public hospitals indicates the frequency of residents going to high-level medical institutions for medical service; The number of consultations per capita in primary medical and health institutions is the same. |
| | | The proportion of consultations in different levels of medical institutions (*dtr*) | The construction of the proportion of public hospitals and the proportion of primary medical and health institutions indicates the tendency of people to choose medical treatment to go to different medical institutions. |
| | Diagnosis and treatment behavior choice | The number of hospitalizations per capita (*htp*) | The per capita hospitalization of medical and health institutions represents the frequency of hospitalization in all medical institutions; the per capita resident of public hospital indicates the frequency of hospitalization in public hospital; The average hospitalization number of permanent residents in primary medical and health institutions is the same. |
| | | The proportion of inpatients in different levels of medical institutions (*htr*) | The proportion of inpatients in public hospitals and the inpatients in primary medical and health institutions indicates the tendency of county people to be hospitalized in different medical institutions. |
| | | Surgery rate in hospitalized patients (*htsr*) | The surgery rate of inpatients in hospitals and public hospitals is used to indicate the behavior tendency of hospital diagnosis and treatment behavior. |
| | Allocation of medical resources | The number of Health workers per thousand people (*pher*) | The number of health personnel per 1,000 permanent residents in hospitals, health personnel per 1,000 rural residents in township health centers, and health personnel per 1,000 urban residents in community health service centers represents the allocation of human resources at various levels of medical institutions. |
| | | The Number of beds per thousand people (*phbed*) | The number of beds per 1,000 permanent residents of medical institutions, primary medical and health institutions, community health service centers, township health centers and outpatient departments indicates the allocation of medical facilities by medical institutions at all levels. |
| Control Variables | The level of population aging (*agingdegree*) | | The proportion of the population over 65 years old in the total population is expressed in units:% |
| | sex structure (*sexrate*) | | The ratio of male to female population represents the local gender structure, unit (%) |
| | Urbanization level (*urpopustr*) | | The ratio of the urban population to the total permanent resident population indicates the local urbanization level, per unit:% |
| | The number of confirmed COVID-19 cases (*covid_19*) | | The number of COVID-19 cases indicates the impact of COVID-19 from the end of 2019 to 2021, unit: person |
| | per capita gross domestic product (*lngdp*) | | Per capita GDP is log taken to indicate the level of local economic development |

Explanatory Variables: These are variables that are used to explain or predict changes in the dependent variables (the outcomes). In the context of the paper, the primary explanatory variable of interest is whether a county was part of the pilot project of CCMC or not.

Dependent Variables: These are the outcomes or variables that the paper aims to study and analyze. The paper looks at various dependent variables, such as the county-level consultation rate, medical service capacity, and resource allocation. The goal is to assess how these variables changed in pilot areas compared to non-pilot areas.

Control Variables: Control variables are used to account for potential confounding factors or other variables that could affect the outcomes. They help ensure that any observed effects

are not due to other factors. The paper used various control variables obtained from the Wind Database.

The paper acknowledges inconsistencies in the subjects of the indicators used. For instance, The Yearbook discloses the number of consultations in Medical and health institutions (Medical and health institutions include hospitals, primary medical institutions, professional public health institutions, and other medical and health institutions), public hospitals, and Primary medical institutions (including community health-service centers, community health-service stations, street health-care centers, township health-care centers, and village health-care centers), but the corresponding subjects for the indicator of health personnel are only hospitals, community health-care centers, and township health-care centers. To address this, the paper analyzed and tested the effects based on the available data in a manner that is consistent with the subject matter of each indicator.

This comprehensive approach to variable selection allows the paper to conduct a thorough analysis of the impact of the CCMC while considering potential confounding factors.

## 4.3. Descriptive statistical analysis

The descriptive statistical analysis presented in Table 4 compares key indicators related to the HDT promoted by the construction of the CCMC. The analysis includes both the experimental group, which consists of counties participating in the pilot project, and the control group, which comprises counties that did not participate in the pilot.

Key findings from the descriptive analysis include significant differences in various health-care-related indicators between the two groups. These indicators include:the number of consultations per capita; the number of hospitalizations per capita; the surgery rate in hospitalized patients; the number of health workers per thousand people; the number of beds per thousand people. These significant differences suggest the possibility of selection bias in the

**Table 4. Descriptive statistical analysis of the pilot project of CCMC.**

| 变量 | Medical institution level | Experimental group | | | Control group | | |
|---|---|---|---|---|---|---|---|
| | | sample capacity | average value | standard deviation | sample capacity | average value | standard deviation |
| *dtp* | Medical and health institutions | 333 | 7.348 | 11.666 | 1313 | 5.627 | 9.952 |
| | Public hospitals | 332 | 2.029 | 2.621 | 1309 | 1.677 | 1.904 |
| | Primary medical institutions | 333 | 5.020 | 9.601 | 1313 | 3.665 | 8.628 |
| *htp* | Medical and health institutions | 333 | 0.254 | 0.411 | 1313 | 0.197 | 0.265 |
| | Public hospitals | 332 | 0.135 | 0.205 | 1309 | 0.106 | 0.131 |
| | Primary medical institutions | 333 | 0.088 | 0.209 | 1313 | 0.063 | 0.137 |
| *htsr* | Hospitals | 444 | 0.239 | 0.132 | 1746 | 0.225 | 0.126 |
| | Public hospitals | 443 | 0.254 | 0.149 | 1738 | 0.235 | 0.131 |
| *pher* | Hospitals | 333 | 517.789 | 750.142 | 1314 | 448.455 | 527.020 |
| | Township health-care centers | 481 | 187.785 | 86.813 | 1823 | 198.775 | 97.419 |
| | Community health-service centers | 396 | 63.884 | 55.230 | 1357 | 64.848 | 61.783 |
| *phbed* | Public hospitals | 333 | 371.538 | 565.627 | 1314 | 305.853 | 329.266 |
| | Community health-service station | 333 | 3.465 | 22.780 | 1314 | 1.508 | 9.185 |
| | Outpatient department | 333 | 1.105 | 7.145 | 1314 | 0.332 | 2.395 |
| *agingdegree* | | 259 | 14.201 | 3.633 | 1022 | 13.416 | 3.862 |
| *sexrate* | | 481 | 1.046 | 0.048 | 1887 | 1.051 | 0.043 |
| *urpopustr* | | 333 | 46.201 | 15.092 | 1314 | 44.203 | 18.918 |
| *covid_19* | | 259 | 20.174 | 88.084 | 1022 | 16.460 | 77.407 |
| *lngdp* | | 481 | 10.408 | 0.616 | 1889 | 10.242 | 0.658 |

determination of pilot areas. In other words, the areas selected for the pilot may have distinctive characteristics. This could lead to endogeneity issues if straightforward Ordinary Least Squares (OLS) estimation is conducted. To address this, it is necessary to first tackle the problem of sample selection bias before evaluating the effects of the pilot project.

## 4.4.Models

The paper employs the PSM-DID model, which combines the Propensity Score Matching (PSM) method and the Double Difference (DID) method. This approach effectively addresses the presence of observable and unobservable confounding factors between the treatment group (medical community pilot areas) and the control group. In the context of the CCMC pilot policy, it essentially represents a quasi-natural experiment. Therefore, conducting a policy evaluation directly through the DID approach can be susceptible to self-selection bias, potentially leading to endogeneity. However, using the PSM method enables the matching of specific control group samples to each treatment group sample. This makes the quasi-natural experiment approximation more like a random one, reducing the impact of self-selection bias and increasing the reliability of the analysis. In essence, PSM-DID is a rigorous approach to evaluate the impact of policies or interventions in a non-randomized setting while minimizing potential biases. It allows for more robust and unbiased assessments of policy effectiveness [55,56].

**4.4.1. Propensity score matching (PSM).** Propensity Score Matching (PSM) is a statistical method used to create a counterfactual framework that helps mitigate the impact of selection bias when estimating parameters for counties that share similar characteristics and have implemented the CCMC pilot project. The form of the binary logit model for estimating the propensity score is as follows:

$$\ln\left(\frac{p_i}{1 - p_i}\right) = \beta_0 + \beta_1 x_i' + u_i \tag{1}$$

Where:$p_i = p(X_i) = Pr(T_i = 1|X_i)$ represents the propensity score of the sample counties, while $x_i'$ signifies the covariates selected through iterative comparison, including population aging level, gender structure, urbanization level, the number of COVID-19 cases, per capita gross domestic product, and resource allocation of medical and health institutions (health personnel per thousand residents). In this study, caliper matching with a radius of 0.05 was employed to retain matched samples while eliminating unmatched ones.

**4.4.2.Double checking (DID).** To further control for potential unobservable variables, this paper incorporates individual and time-fixed effects. Using panel data, the influence of county-level compact medical community construction on residents' choice of medical treatment can be explored using the following formulas:

$$y_{it}^{1s} = \beta \times c_i + \theta_i + x_{it}'\gamma + \tau_t + \epsilon_{it} \tag{2}$$

$y_{it}^{1s}$ represents the number of consultations per capita to s medical institutions at time t in county $i$. $c_i$ is a binary variable, taking the value 1 when the county undergo the CCMC, and 0 otherwise. $\theta_i$ stands for the fixed effect of the county, controlling for heterogeneity at the county level. $x_{it}'$ represents a set of control variables including the level of population aging, gender ratio urbanization level, COVID-19 cases, per capita gross domestic product, and resource allocation of medical and health institutions (health personnel per 1,000 permanent population). $\tau_t$ represents the fixed effect of the year, controlling for the common time impact of cities and states in different periods. $\epsilon_{it}$ is the residual, and robust standard error is employed

for estimation.

$$y_{it}^{2s} \, or \, y_{it}^{3s} = \beta \times c_i + \theta_i + x_{it}'\gamma + \tau_t + \epsilon_{it} \tag{3}$$

$y_{it}^{2s}$ represents the number of hospitalizations per capita of county $i$ in $s$ medical institutions with time $t$, and $y_{it}^{3s}$ represents the rate of inpatient surgery of county $i$ in $s$ medical institutions at time $t$. Other indicators are the same as formula (2).

Finally, how and the construction of medical community affects the resource allocation of medical institutions. The formula is as follows:

$$y_{it}^{4s} \, or \, y_{it}^{5s} = \beta \times c_i + \theta_i + x_{it}'\gamma + \tau_t + \epsilon_{it} \tag{4}$$

$y_{it}^{4s}$ represents the health personnel per thousand permanent population in s medical institutions at time $t$. $y_{it}^{5s}$ represents the number of beds per thousand permanent resident population in s medical institutions at time $t$. Other indicators are the same as formula (2).

## 5.Empirical regression results

### 5.1. Main effect

**5.1.1 Diversion effect.** Table 5 illustrates the impact of CCMC pilot project in Sichuan Province under the package payment of medical insurance on guiding patients towards medical shunting. In general, the number of consultations per capita of medical and health institutions in the pilot counties of the medical community increased significantly, by 0.434 times. When looking at medical institutions of different levels, it is observed that the number of consultations per capita to public hospitals in the pilot areas and counties did not change significantly. However, the number of consultations per capita to basic medical institutions increased significantly, by 0.340 times.

While it might appear that there hasn't been a significant change in the proportion of consultations between public hospitals and primary medical institutions, it is important to note that this balance has attracted the return of severe patients and encouraged medical shunting at the grassroots level. This effect seems to have somewhat canceled each other out in the structure. The results suggest that the county-level consultation rate has improved following the

**Table 5. Influence of county health community construction on patients' medical shunt.**

| Explanatory variable | Dependent Variables | | | | |
|---|---|---|---|---|---|
| | The number of consultations per capita | | | The proportion of consultations | |
| | Medical and health institutions | Public hospitals | Primary medical institutions | Public hospitals | Primary medical institutions |
| ifygt | 0.434** | 0.085 | 0.340** | 0.004 | -0.004 |
| | (0.185) | (0.088) | (0.134) | (0.011) | (0.011) |
| Constant | -5.559 | -2.320 | -2.714 | 0.468 | 0.501* |
| | (4.971) | (2.222) | (3.674) | (0.318) | (0.293) |
| Annual effect | yes | yes | yes | yes | yes |
| Individual effect | yes | yes | yes | yes | yes |
| *R-squared* | 0.901 | 0.949 | 0.883 | 0.945 | 0.951 |
| Sample capacity | 390 | 387 | 390 | 390 | 387 |

Robust standard errors in parentheses,

*** p<0.01,

** p<0.05,

* p<0.1.

**Table 6. Influence of CCMC construction on diagnosis and treatment behavior of medical institutions at all levels.**

| Explanatory variable | Dependent Variables | | | | | | |
|---|---|---|---|---|---|---|---|
| | The number of hospitalizations per capita | | | The proportion of inpatients | | Surgery rate in hospitalized patients | |
| | Medical and health institutions | Public hospitals | Primary medical institutions | Public hospitals | Primary medical institutions | Hospitals | Public hospitals |
| ifygt | 0.020*** | 0.011*** | 0.009*** | 0.007 | 0.006 | 0.044** | 0.050** |
| | (0.007) | (0.004) | (0.003) | (0.009) | (0.013) | (0.020) | (0.023) |
| Constant | -0.329* | -0.067 | -0.176** | -0.288 | 1.302*** | 2.411*** | 2.651*** |
| | (0.188) | (0.107) | (0.088) | (0.249) | (0.327) | (0.667) | (0.790) |
| Annual effect | yes | yes | yes | yes | yes | yes | yes |
| Individual effect | yes | yes | yes | yes | yes | yes | yes |
| R-squared | 0.922 | 0.955 | 0.916 | 0.969 | 0.964 | 0.708 | 0.664 |
| Sample capacity | 390 | 387 | 390 | 390 | 387 | 390 | 387 |

Robust standard errors in parentheses,

*** p<0.01,

** p<0.05,

* p<0.1.

pilot project of CCMC, and the pilot project has indeed played a role in guiding county patients to seek medical care locally and promoting initial diagnoses at the grassroots level to a certain extent.

**5.1.2.Diagnosis and treatment effect.** Table 6 provides insights into the impact of CCMC pilot in Sichuan Province on guiding the treatment behavior of medical institutions at all levels under the package payment of medical insurance. Notably, the total number of hospitalizations per capita to medical and health institutions saw a significant increase of 0.02 (According to the descriptive statistical analysis, the number of hospitalizations per capita in medical and health institutions in the experimental and control groups was 0.254 and 0.197, respectively).

When examining the number of hospitalizations per capita to public hospitals and grassroots medical institutions, it becomes evident that the pilot of CCMC mainly led to an increase in the number of hospitalizations per capita to public hospitals. However, the proportion of inpatient in public hospitals and grassroots medical institutions in the pilot counties did not undergo significant changes.

Furthermore, the total inpatient surgery rate of hospitals and the inpatient surgery rate of public hospitals increased by 4.4% and 5%, respectively. This indicates that the pilot has not only encouraged initial diagnoses at the grassroots level but has also enhanced the medical service capabilities of county-level medical and health institutions. It has also promoted public hospitals to resume their role in dealing primarily with complex and severe diseases, thereby further optimizing the healthcare pattern of "major diseases being addressed in hospitals, and minor diseases being managed in the grassroots medical institutions."

**5.1.3.Resource allocation effect.** Table 7 illustrates the impact of the CCMC pilot project in Sichuan province on guiding the behavior of medical institutions at all levels in diagnosis and treatment under the package payment method of medical insurance. Several key observations can be made from the data: There was no significant change in the number of health personnel per 1,000 population in medical institutions at all levels. This suggests that the pilot project has not significantly affected the allocation of health human resources. The number of beds per 1,000 population in community health service stations increased significantly by

**Table 7. Influence of CCMC construction on the allocation of medical resources.**

| Explanatory variable | Dependent Variables | | | | | | | |
|---|---|---|---|---|---|---|---|---|
| | Health workers per thousand people | | | The number of beds per thousand people | | | | |
| | Hospitals | Township health-care centers | Community health-service centers | Public hospitals | Community health-service centers | Community health-service station | Township health-care centers | Outpatient department |
| ifygt | -15.665 | -18.045 | -1.471 | -3.363 | 0.769 | 1.678** | 0.785 | -0.739* |
| | (14.558) | (11.964) | (5.976) | (13.646) | (1.958) | (0.760) | (9.914) | (0.436) |
| Constant | -721.077** | 730.793 | 105.614 | -56.426 | 21.515 | -1.753 | -223.346 | 19.425** |
| | (354.627) | (473.758) | (177.488) | (284.466) | (32.632) | (22.232) | (181.756) | (9.335) |
| Annual effect | yes | yes | yes | yes | yes | yes | yes | yes |
| Individual effect | yes | yes | yes | yes | yes | yes | yes | yes |
| R-squared | 0.965 | 0.901 | 0.851 | 0.928 | 0.729 | 0.834 | 0.770 | 0.475 |
| Sample capacity | 392 | 360 | 304 | 392 | 392 | 392 | 392 | 392 |

Robust standard errors in parentheses,

*** $p < 0.01$,

** $p < 0.05$,

* $p < 0.1$.

1.678. This indicates that the pilot project has led to an increase in the allocation of medical facility resources at the community level. Conversely, the number of beds per 1,000 population in outpatient departments of hospitals decreased significantly by 0.739. This decrease may reflect a shift in resource allocation, potentially indicating that medical resources are being redirected away from hospital outpatient departments. In summary, the pilot project has had a mixed impact on the allocation of resources, with health personnel remaining largely unchanged, but with notable changes in the allocation of medical facility resources.

## 5.2.Parallel trend test

The DID model relies on the parallel trend hypothesis, which assumes that, before implementing the policy, cities that implement outpatient coordination and those that do not should exhibit parallel trends in various factors, including the number of medical institutions at all levels, the outpatient and emergency hospitalization rates of public hospitals, and the allocation of medical resources in public hospitals. To test this hypothesis, the paper employs the event study method proposed by Jacobson et al (1993) [57] for a parallel trend analysis.

The analysis focuses on the main regression results related to the three effects mentioned above. It compares the data in the pre-implementation phase (pre3) to the data in the first three phases of the pilot project. The results of the parallel trend test in Tables 8 and 9 reveal that, except for the number of beds per 1,000 population in hospital outpatient departments, the coefficients of the variables analyzed in this paper are not statistically significant in the pre-implementation period (pre3, pre2). However, they become significant in the current and subsequent periods (current, post1, post2) following the implementation of the pilot policy.

In order to avoid issues related to multicollinearity, the baseline group was defined as the first period before the policy (pre1) and was excluded from the analysis. This suggests that, prior to the policy implementation, there were no significant differences in the number of outpatient visits, inpatient admissions, inpatient surgery rates, the number of health workers per 1,000 population, and the number of beds per 1,000 population at medical institutions of all

**Table 8. Parallel trend test of diversion effect and resource allocation effect.**

| Explanatory variable | Dependent Variables | | | |
|---|---|---|---|---|
| | The number of consultations per capita | | The number of beds per thousand people | |
| | Medical and health institutions | Primary medical institutions | Community health service station | Outpatient department |
| pre_3 | 0.089 | 0.117 | 1.122 | -0.739 |
| | (0.359) | (0.284) | (1.154) | (0.601) |
| pre_2 | -0.070 | -0.065 | 0.961 | -0.318 |
| | (0.306) | (0.213) | (1.217) | (0.457) |
| current | 0.220 | 0.214 | 3.473** | -0.915 |
| | (0.284) | (0.198) | (1.441) | (0.646) |
| post_1 | 0.396 | 0.350* | 1.541* | -0.954 |
| | (0.290) | (0.211) | (0.914) | (0.708) |
| post_2 | 0.661** | 0.456** | 1.884* | -1.186 |
| | (0.278) | (0.218) | (0.985) | (0.809) |
| Constant | -5.274 | -2.485 | 0.038 | 19.330** |
| | (4.965) | (3.662) | (22.665) | (9.426) |
| Annual effect | yes | yes | yes | yes |
| Individual effect | yes | yes | yes | yes |
| R-squared | 0.902 | 0.884 | 0.836 | 0.477 |
| Sample capacity | 390 | 390 | 392 | 392 |

Robust standard errors in parentheses,

*** $p < 0.01$,

** $p < 0.05$,

* $p < 0.1$.

**Table 9. Parallel trend test of diagnosis and treatment effect.**

| Explanatory variable | Dependent Variables | | | | |
|---|---|---|---|---|---|
| | The number of hospitalizations per capita | | | Surgery rate in hospitalized patients | |
| | Medical and health institutions | Public hospitals | Primary medical institutions | Hospitals | Public hospitals |
| pre_3 | 0.001 | -0.000 | -0.001 | 0.007 | -0.008 |
| | (0.011) | (0.007) | (0.004) | (0.023) | (0.027) |
| pre_2 | -0.001 | 0.001 | -0.002 | 0.009 | -0.000 |
| | (0.011) | (0.007) | (0.005) | (0.019) | (0.023) |
| current | 0.014 | 0.008 | 0.008* | 0.045 | 0.040 |
| | (0.010) | (0.006) | (0.004) | (0.041) | (0.049) |
| post_1 | 0.020* | 0.012* | 0.007 | 0.046** | 0.046* |
| | (0.011) | (0.007) | (0.004) | (0.021) | (0.024) |
| post_2 | 0.025** | 0.013** | 0.011* | 0.053* | 0.058* |
| | (0.011) | (0.006) | (0.006) | (0.028) | (0.034) |
| Constant | -0.322* | -0.065 | -0.176** | 2.421*** | 2.651*** |
| | (0.190) | (0.108) | (0.088) | (0.660) | (0.782) |
| Annual effect | yes | yes | yes | yes | yes |
| Individual effect | yes | yes | yes | yes | yes |
| R-squared | 0.923 | 0.956 | 0.916 | 0.709 | 0.664 |
| Sample capacity | 390 | 387 | 390 | 390 | 387 |

Robust standard errors in parentheses,

*** $p < 0.01$,

** $p < 0.05$,

* $p < 0.1$.

levels in different cities and states implementing outpatient coordination at different times. This indicates that other factors did not significantly influence these factors. In essence, the study samples passed the parallel trend test.

### 5.3. Placebo test

To further ensure the robustness of the results, and exclude the influence of other policy changes or random factors, this paper conducted a placebo test using a fictional policy time frame and a fictional treatment group. In this test, the integration of urban and rural residents' medical insurance in 2016 might affect the dependent variables of interest in this study. Following the approach of Liu and Zhao (2015) [58], the pilot timeframe was moved two years ahead to 2017. The results showed that, with the exception of a few variables with weak significance, the estimated results of the dependent variables were not significant (See all variables in S3 File).

Following the approach of Cai et al (2016) [59], 37 counties were randomly selected from the 183 sample counties to serve as the fictitious treatment group cities. The remaining counties were designated as the fictitious control group cities. This was done to obtain coefficient estimates for the impact of the pilot project on the public's medical treatment triage, hospital diagnosis and treatment behavior, and medical resource allocation.

The entire process was repeated 500 times, resulting in 500 sets of regression coefficients and their corresponding P-values. Fig 2 illustrates the kernel density distribution and P-value distribution of part variables (See all variables in S1 File). By visualizing the kernel density distribution and P-value distribution of these 500 coefficients, it becomes apparent that the regression coefficients are centered around zero and follow a normal distribution except for the number of beds in the community health service station. The majority of regression results are not statistically significant. The main effect regression results are positioned at the tail end of the positive distribution, representing rare occurrences. Therefore, it can be concluded that the main effect results presented in this paper are not due to confounding factors.

### 5.4. Robustness test

To ensure the robustness of the main regression results, a robustness test was performed by adjusting the data years. The analysis was conducted using data from 2016 to 2021. The estimated coefficients and their significance obtained from the results presented in Tables 10 and 11 were found to be consistent with the original sample results. This indicates that the main regression results are robust and hold across different data years.

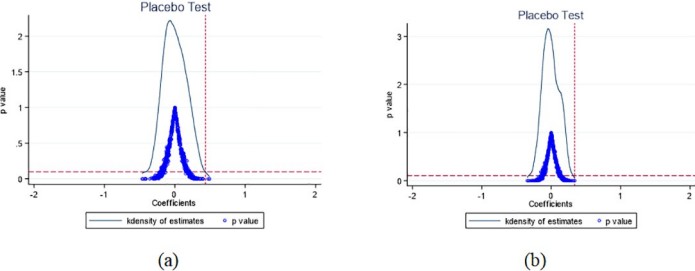

**Fig 2. Placebo test of the number of consultations per capita of Medical and health institutions(a)and Primary medical institutions(b).**

**Table 10. Robustness test of diversion effect and resource allocation effect in the medical community pilot.**

| Explanatory variable | Dependent Variables | |
|---|---|---|
| | The number of consultations per capita | |
| | Medical and health institutions | Primary medical institutions |
| ifygt | 0.335* | 0.279* |
| | (0.199) | (0.148) |
| Constant | -5.426 | -2.294 |
| | (5.884) | (4.245) |
| Annual effect | yes | yes |
| Individual effect | yes | yes |
| R-squared | 0.910 | 0.891 |
| Sample capacity | 321 | 321 |

Robust standard errors in parentheses,

\*\*\* p<0.01,

\*\* p<0.05,

\* p<0.1.

## 6.Discussion

Based on the data of Sichuan Health Statistical Yearbook from 2008 to 2021, this paper uses the PSM-DID method to evaluate the pilot effectiveness of the CCMC in Sichuan Province.

The main reason for the low willingness of residents to seek medical treatment at the grass-roots level is the insufficient capacity of primary medical services and insufficient equipment, resulting in insufficient trust in the first diagnosis at the grass-roots level [8]. Under the policy guidance, grassroots medical institutions pay more attention to public health than medical care, and grassroots medical institutions have weak performance incentives. The compensation mechanism is incomplete, and the salary system and management system are backward, resulting in insufficient motivation of grassroots medical personnel [11–13].

Through the reform of medical insurance payment mechanism, personnel performance appraisal and preparation mechanism, family doctor signing mechanism and information

**Table 11. Robustness test of the diagnosis and treatment effect in the medical community trial.**

| Explanatory variable | Dependent Variables | | | | |
|---|---|---|---|---|---|
| | The number of hospitalizations per capita | | | Surgery rate in hospitalized patients | |
| | Medical and health institutions | Public hospitals | Primary medical institutions | Hospitals | Public hospitals |
| ifygt | 0.017** | 0.009* | 0.009** | 0.041** | 0.051** |
| | (0.007) | (0.004) | (0.004) | (0.020) | (0.024) |
| Constant | -0.163 | -0.059 | -0.104 | 2.455*** | 2.786*** |
| | (0.217) | (0.154) | (0.102) | (0.773) | (0.916) |
| Annual effect | yes | yes | yes | yes | yes |
| Individual effect | yes | yes | yes | yes | yes |
| R-squared | 0.929 | 0.960 | 0.925 | 0.734 | 0.694 |
| Sample capacity | 321 | 321 | 321 | 321 | 319 |

Robust standard errors in parentheses,

\*\*\* p<0.01,

\*\* p<0.05,

\* p<0.1.

sharing mechanism, Sichuan Province tries to optimize the multi-level principal-agent relationship of "patient-medical insurance-hospital-doctor" in the construction of CCMC through various policy means and systems [2], effectively "bind" the interests of all parties' principals and agents [15], and drive the individual system subject to pursue the maximization of interests in the process of pursuing the maximization of interests, which is consistent with the goal of maximizing collective value [25,26].

From the number of consultations per capita, the number of hospitalizations per capita of medical institutions at all levels before and after the pilot of the CCMC in Sichuan Province, the rate of primary treatment at the county level has been improved after the pilot of CCMC. To a certain extent, the pilot of CCMC has realized the role of guiding the return of patients in the county and promoting the first diagnosis at the grassroots level. The increase in the rate of medical treatment in the county and the rate of primary medical treatment is consistent with the evaluation of the achievements of CCMC in Chengdu, Chongqing and Shanghai [31–33].

The results of the diagnosis and treatment behavior of medical institutions at all levels guided by the county-level compact medical community in Sichuan Province showed that the total surgery rate of inpatients in hospitals and the surgery rate of inpatients in public hospitals increased by 4.4% and 5% respectively. This shows that the pilot encourages public hospitals to return to the function of mainly dealing with difficult and severe diseases, and the medical pattern of serious diseases entering hospitals and minor diseases entering communities has been further optimized, which is consistent with the survey results of Tianchang, Anhui and Naqu, Tibet [34,35].

Although the CCMC in Sichuan province has the effect of HDT, the current effect is still relatively weak especially in promoting the allocation of medical personnel and medical facilities in primary medical organizations. This aspect is very insufficient compared with the construction of the CCMC in Chongqing and Shanghai [32,33]. In the process of building the future medical community, the policymaker must first pay attention to the top-level design of the policy, followed by the systematic design and support of the relevant supporting policies [38,39]. Avoid the failure of resource allocation in the process of organizational integration, resulting in the resources of weaker grassroots institutions being further ' siphoned ' by higher-level institutions [51]; secondly, establish an effective incentive compatibility design to enhance the motivation of collaborators to participate in the alliance and the motivation of resource sharing [40,41].

The study innovatively utilizes the quasi-natural experiment of the pilot of CCMC in Sichuan Province, China, offering a unique approach to evaluate the impact of health policy interventions. The analysis of county-level data from the 'Sichuan Provincial Health Statistics Yearbook' over a span of 13 years (2008–2021) provides a robust and comprehensive dataset for evaluation. This study specifically focuses on HDT in the context of the medical insurance package payment model in China, a relatively new area of research in healthcare policy.

By showing the effectiveness of CCMC in promoting HDT, the study highlights a pathway to improve healthcare accessibility and quality at the county level. The findings provide valuable insights for healthcare policy, especially in developing and implementing effective strategies for HDT in county-level medical institutions.

## 7.Conclusions and policy recommendations

### 7.1. Conclusion

The conclusions drawn in this paper primarily focus on the impact of the pilot implementation of the CCMC in Sichuan Province on the HDT under the package payment method of medical insurance. The main findings are as follows: The pilot project significantly increased the

county consultation rate among residents in the pilot areas, encouraging them to seek primary medical care and actively promoting first-level diagnosis and treatment. The pilot led to an improvement in the hospitalization rate of medical institutions at all levels in the pilot areas, particularly in public hospitals at the county level. Additionally, the pilot project enhanced the inpatient surgery rate in hospitals, indicating improved county-level medical service capacity and the redirection of hospitals to focus on treating severe and complex cases. The pilot failed to promote the sinking of medical infrastructure and human resources. These findings shed light on the effects of the pilot project on healthcare delivery and resource allocation under the framework of medical insurance package payment. The above conclusions provide guidance for the further improvement of the CCMC system: in particular, the acceleration of the CCMC pilot policy to reach more people; the strengthening of financial subsidies for primary health care resource allocation; and the improvement of incentives for the remuneration and staffing of health personnel.

## 7.2. Policy suggestions

**7.2.1.Strengthening the promotion of CCMC pilot policies.** This paper has demonstrated that the pilot implementation of the CCMC in Sichuan Province has effectively guided county patients to seek medical treatment and encouraged first-level diagnosis at grassroots healthcare facilities. These outcomes represent crucial steps towards the gradual realization of HDT within the CCMC. They also provide the foundation for a positive feedback loop within the four-way incentive compatibility mechanism involving the government, medical community, physicians, and patients. To further enhance the success of the county medical system, it is imperative to bolster public awareness and understanding of the medical community pilot policies. Achieving this can be facilitated through differential medical insurance payment policies, the enhancement of the "family doctor" contract system, and the continuous improvement of medical service capacity within the medical community. These efforts will instill greater confidence among county patients in the county's healthcare system.

**7.2.2. Enhancing financial subsidies for primary medical resource allocation.** In terms of the number of beds per 1,000 people, only the number of beds in community health service stations has increased significantly after the medical community pilot but has not passed the placebo test, and there is no significant change in the bed allocation of other primary medical institutions such as township health centers and community health service centers. It can be seen that the current reform of the medical community has not yet played a big role in the allocation of medical infrastructure, especially the resources of rural medical facilities. Grassroots medical infrastructure is also an important factor to attract talents and superior medical personnel to the grass-roots level. Therefore, the allocation of primary medical resources not only requires the overall grasp of the medical community, but also requires the financial departments and health departments to improve the grassroots health infrastructure and resource allocation subsidies.

**7.2.3.Enhancing incentive measures for health personnel in terms of salary and staffing.** Based on the analysis in this paper, the construction of the CCMC in Sichuan Province has not effectively influenced the allocation of medical human resources. One possible explanation for this is that the medical community has not made substantial changes to the establishment of positions to encourage the flow of higher-level health personnel, thereby failing to significantly impact the allocation of healthcare human resources. To address this, it is essential to further refine the salary incentive mechanism and staffing arrangements for healthcare professionals. These adjustments will be instrumental in attracting medical talent to participate

in the development of county-level compact medical communities, especially in building the capacity of primary healthcare services.

## 8.Limitations and research outlook

The limitations of this paper are the limited geographical focus and the lack of other useful data. The study is geographically limited to Sichuan Province, which may affect the generalizability of the findings to other regions or countries. The study primarily relies on quantitative data, potentially overlooking qualitative aspects such as patient satisfaction or the subjective experiences of healthcare providers. The Sichuan Health Statistics Yearbook lacks data on medical costs and utilization of the health insurance fund, so it is not possible to test the impact of the Community on health expenditures and health insurance expenditures.

Further research directions in the future are as follows: expansion to other regions,long-term impact analysis and integration with other health reforms. Future research could explore the implementation and effects of CCMC in different provinces or countries to validate and expand upon these findings. Investigating the long-term impacts of CCMC on healthcare outcomes, cost-effectiveness, and patient satisfaction would be valuable. And examining how CCMC integrates with other health reforms and policies in China and its synergistic effects could provide a more holistic understanding of healthcare improvements.

## Supporting information

**S1 File. Placebo test.**
(DOCX)

**S2 File. Data sample.**
(XLSX)

**S3 File. Code.**
(DOCX)

## Author Contributions

**Conceptualization:** Shaoqun Ding.

**Methodology:** Yuxuan Zhou.

**Software:** Yuxuan Zhou.

**Supervision:** Shaoqun Ding.

**Writing – original draft:** Yuxuan Zhou.

**Writing – review & editing:** Yuxuan Zhou.

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
