## [Decision Letter · Decision Letter 0]

18 Dec 2023

PONE-D-23-37259County Medical Community, Medical Insurance Package Payment, and Hierarchical Diagnosis and TreatmentPLOS ONE

Dear Dr. ZHOU,

Thank you for submitting your manuscript to PLOS ONE. After careful consideration, we feel that it has merit but does not fully meet PLOS ONE’s publication criteria as it currently stands. Therefore, we invite you to submit a revised version of the manuscript that addresses the points raised during the review process.

We look forward to receiving your revised manuscript.

Kind regards,

Alfredo Luis Fort, M.D., M.Sc., Ph.D.

Academic Editor

PLOS ONE

Journal Requirements:

National Social Science Fund Major Project ' Research on Multi-level Social Security System Innovation and Policy Synergy Based on System Concept ' ( 23ZDA099 ) ; the basic scientific research business fee of the central university ' Research on the effect of urban and rural residents ' medical insurance outpatient co-ordination to promote hierarchical diagnosis and treatment ' graduate research project ( JBK2307025 ). 

Additional Editor Comments:

Dear Author: You will see the comments from two reviewers, who conclude your manuscript needs to undergo a minor revision, adding and clarifying things. I have also included several suggestions in my attached file, for your to revise and improve descriptions and statements, for the benefit of the study and the reader.

IMPORTANT SUGGESTION: You will see one reviewer has suggested you add several studies. PLEASE, DO NOT FEEL OBLIGED TO ADD THESE STUDIES, unless you see they are relevant and/or important for your manuscript. Thanks.

Reviewers' comments:

Reviewer's Responses to Questions

**Comments to the Author**

1. Is the manuscript technically sound, and do the data support the conclusions?

Reviewer #1: Yes

Reviewer #2: Yes

2. Has the statistical analysis been performed appropriately and rigorously? 

Reviewer #1: Yes

Reviewer #2: Yes

3. Have the authors made all data underlying the findings in their manuscript fully available?

Reviewer #1: Yes

Reviewer #2: Yes

4. Is the manuscript presented in an intelligible fashion and written in standard English?

Reviewer #1: Yes

Reviewer #2: Yes

5. Review Comments to the Author

Reviewer #1: I reviewed this work and found it interesting and have merits. However, a couple of sticky notes added which should be incorporated in revised version

a) A couple of pieces of literature should be added to the revised version related to medical or health expenditures and their impact.

Abbas, H. S. M., Azhar, A., Gul, A.A. (2024). Impact of Environmental and Economic determinants on Life Expectancy: A Sustainable Development Exploration in Sino-Pak from 1965 to 2020. Int. J. of Environment and Sustainable Development, 23 (1). Doi: 10.1504/IJESD.2022.10052100

Abbas, H. S. M., Xu, X., Sun, C., & Abbas, S. (2023). Impact of administrative state capacity determinants on sustainable healthcare. Heliyon, 9(7). Doi: 10.1016/j.heliyon.2023.e18273

Abbas, H. S. M., Xu, X., & Sun, C. (2022). The role of state capacity and socio-economic determinants on health quality and its access in Pakistan (1990–2019). Socio-Economic Planning Sciences, 83, 101109. Doi: 10.1016/j.seps.2021.101109

b) The study's innovation and significance should be highlighted and explained clearly in the discussion.

c) Add study limitations and future study directions.

d) Minor language editing should also be required to polish this work before publication.

Good Luck

Reviewer #2: Title – County Medical Community, Medical Insurance Package Payment, and Hierarchical

Diagnosis and Treatment

Abstract - Well written but need to include key policy implication in the concluding part of the abstract

Background –

Detailed information on statement of problem as well as rational for the study clearly presented. The study objective stated appropriately

Methods

Well described in details. The data management is well prepared.

Result

Well written in details with relevant tables.

Discussion

There is no discussion on this manuscript. Findings / Result of this study were not discussed at all. The findings were not explained and not compared with findings from other studies.

Conclusion –

Clearly written but without very clear policy implication and recommendation.

6. PLOS authors have the option to publish the peer review history of their article (what does this mean?). If published, this will include your full peer review and any attached files.

Reviewer #1: No

Reviewer #2: **Yes: **Prof. Tanimola Makanjuola Akande

---

## [Author Response · Author response to Decision Letter 0]

26 Dec 2023

Subject: Response to academic editor's Comments on Manuscript PONE-D-23-37259

Dear Dr. Fort,

We appreciate your detailed and helpful revisions regarding our manuscript titled "County Medical Community, Medical Insurance Package Payment, and Hierarchical Diagnosis and Treatment" with the ID PONE-D-23-37259. We appreciate the opportunity to revise our manuscript and have addressed each of the points you raised in your review. Below, I provide a detailed response to each of the points:

Journal Requirements:

1.Comment on Style Requirements:

Editor's Comment: "Please ensure that your manuscript meets PLOS ONE's style requirements, including those for file naming."

Response: We have thoroughly reviewed the manuscript and ensured that it now fully complies with PLOS ONE's style requirements. This includes adhering to the guidelines for file naming. We have renamed the files accordingly and rechecked the manuscript format for complete compliance.

2.Comment on Code Sharing Guidelines:

Editor's Comment: "PLOS ONE has specific guidelines on code sharing for submissions in which author-generated code underpins the findings in the manuscript. All author-generated code must be made available without restrictions upon publication."

Response: In accordance with PLOS ONE's code sharing guidelines, we have prepared our author-generated code for sharing. We will make it available without restrictions upon the publication of our work. The relevant data and code named “S4_File. Data sample. (dta)” and “S5_File.code. (do)” has been uploaded with the Manuscript.

3.Comment on Role of Funders:

Editor's Comment: "Please state what role the funders took in the study, or state that 'The funders had no role in study design, data collection and analysis, decision to publish, or preparation of the manuscript.' Include this in your cover letter."

Response: We have included a revised 'Role of Funder' statement in our cover letter as requested. In our study, the funders had no role in study design, data collection and analysis, decision to publish, or preparation of the manuscript.

4.Comment on Reference List:

Editor's Comment: "Please review your reference list to ensure that it is complete and correct."

Response: We have carefully reviewed and revised our reference list to ensure its completeness and accuracy. All references have been double-checked for correctness and adherence to the PLOS ONE citation format.

Revisions of Manuscript

1.Comment on Ethics Statement:

Editor's Comment: "One has to put something at the Ethics Statement, because one is dealing with patients seen by the medical establishment..."

Response: We have added an Ethics Statement to our manuscript. Although our study uses secondary data with no specific personal data. Here are the specifics of the ethics statement: this paper discussed the Diagnosis and treatment of patients seen by the medical establishment using secondary data from the Sichuan Health Statistics Yearbook and there is no information on specific personal data.

2.Comment on Defining Key Terms:

Editor's Comment: "Some important noun needs to be better defined and termed..."

Response: We have reviewed our manuscript and provided clearer definitions and terms for key concepts, such as Hierarchical diagnosis and treatment (HDT), compact county medical communities (CCMC), and Pareto improvements. We believe these revisions will enhance readers' understanding of our study.

3.Comment on Inconsistent Definitions:

Editor's Comment: "Some definition has nothing to do with the previous column in terms of variable settings"

Response: We apologize for the oversight and have rectified the inconsistency in definitions between columns in terms of variable settings. The revised manuscript now ensures coherence in the presentation of all terms and concepts.

4.Comment on Clarity in Tables:

Editor's Comment: "Try putting the full term in tables, otherwise, the reader has to go back and look at what this means..."

Response: We have revised our tables to include full terms where necessary, enhancing readability and ensuring that readers do not need to refer back constantly.

5.Comment on Placebo Test Section:

Editor's Comment: "Perhaps add at the Placebo test section that the data are available for any interested reader."

Response: In the Placebo test section, we have now stated that the data used in our study are available to interested readers upon request, enhancing the transparency of our research.

6.Comment on Discussing Structural Changes:

Editor's Comment: "It may be important to discuss that more 'structural' changes such as number of beds, plus the allocation of human resources like doctors to other areas are more COMPLEX operations..."

Response: We have added a comprehensive discussion section where we interpret our findings, compare them with existing literature, and explore their broader implications. This new section can be found in page 23-24.

Many of the minor formatting issues in the revisions have been revised, as detailed in the Manuscript, and are not repeated here.

We hope that the revisions and clarifications made in our manuscript now fully address these concerns. We are confident that these changes have substantially improved the quality of our work. We thank you for the constructive feedback, which has undoubtedly strengthened our manuscript.

We look forward to the revised manuscript being considered for publication in PLOS ONE and are enthusiastic about the potential contribution our study offers to the field.

Happy New Year 2024 in Advance!

Sincerely,

Yuxuan Zhou

School of Public Administration, Southwestern University of Finance and Economics, Chengdu, Sichuan, China

Liulin Campus, Southwestern University of Finance and Economics, 555 Liutai Avenue, Wenjiang District, Chengdu, Sichuan, China

E-mail: 15556932867@163.com (Zhou)

On December 26, 2023

Subject: Response to Reviewer #1's Comments on Manuscript PONE-D-23-37259

Dear Reviewer #1,

Thank you for sharing the insightful comments regarding our manuscript titled "County Medical Community, Medical Insurance Package Payment, and Hierarchical Diagnosis and Treatment" with the ID PONE-D-23-37259. We are grateful for the positive feedback and have carefully addressed each of these concerns in the revised version of our manuscript. Below, I outline our responses to each of the points raised:

1.Comment on Literature Addition:

Reviewer's Comment: "A couple of pieces of literature should be added to the revised version related to medical or health expenditures and their impact."

Response: We appreciate the reviewer's suggestion to enrich our manuscript with additional literature. Accordingly, we have incorporated several relevant studies that discuss the impact of medical and health expenditures. These can be found in page 6（References [19] and [20] ）.

2.Comment on Innovation and Significance:

Reviewer's Comment: "The study's innovation and significance should be highlighted and explained clearly in the discussion."

Response: We agree with highlighting the study's innovation and significance is crucial. We have added the discussion section to clearly articulate these aspects, emphasizing how our study contributes uniquely to the field. Please refer to page 24, the last two paragraphs of the discussion section for these detailed explanations.

3.Comment on Study Limitations and Future Directions:

Reviewer's Comment: "Add study limitations and future study directions."

Response: Recognizing the importance of acknowledging our study's limitations and suggesting future research directions, we have added a section in the manuscript that addresses these points. This addition not only provides transparency regarding our study's scope but also suggests avenues for future research. This information is now included in page 26, final section of the paper: Limitations and research outlook.

4.Comment on Language Editing:

Reviewer's Comment: "Minor language editing should also be required to polish this work before publication."

Response: We have thoroughly reviewed the manuscript and made comprehensive language edits to enhance clarity and readability. We also sought assistance from Professional to ensure the quality of our manuscript's language.

We hope that the revisions and clarifications made in our manuscript now fully address these concerns. We are confident that these changes have substantially improved the quality of our work. We thank you for the constructive feedback, which has undoubtedly strengthened our manuscript.

We look forward to the revised manuscript being considered for publication in PLOS ONE and are enthusiastic about the potential contribution our study offers to the field.

Happy New Year 2024 in Advance!

Sincerely,

Yuxuan Zhou

School of Public Administration, Southwestern University of Finance and Economics, Chengdu, Sichuan, China

Liulin Campus, Southwestern University of Finance and Economics, 555 Liutai Avenue, Wenjiang District, Chengdu, Sichuan, China

E-mail: 15556932867@163.com (Zhou)

On December 26, 2023

Subject: Response to Reviewer #2's Comments on Manuscript PONE-D-23-37259

Dear Prof. Akande,

Thank you for sharing the insightful comments regarding our manuscript titled "County Medical Community, Medical Insurance Package Payment, and Hierarchical Diagnosis and Treatment" with the ID PONE-D-23-37259. We appreciate the positive feedback on several sections of our paper and have taken great care to address the concerns raised. Below, I provide a detailed response to each of your comments:

1.Comment on Abstract's Conclusion:

Reviewer's Comment: "Well written but need to include key policy implication in the concluding part of the abstract."

Response: We agree with the reviewer that the policy implications are crucial in the abstract. We have revised the abstract to include a concise statement on the key policy implications derived from our study. This revision can be found at the end of the abstract in page 2.

2.Comment on the Lack of Discussion:

Reviewer's Comment: "There is no discussion on this manuscript. Findings / Result of this study were not discussed at all. The findings were not explained and not compared with findings from other studies."

Response: We acknowledge the reviewer’s concern regarding the absence of a discussion section. To rectify this, we have added a comprehensive discussion section where we interpret our findings, compare them with existing literature, and explore their broader implications. This new section can be found in page 23-24.

3.Comment on Conclusion and Policy Implications:

Reviewer's Comment: "Clearly written but without very clear policy implication and recommendation."

Response: We appreciate the reviewer's emphasis on the importance of policy implications and recommendations in the conclusion. Accordingly, we have revised the conclusion to more explicitly state the policy implications and recommendations stemming from our research. These additions are now clearly articulated in page 25, the concluding section.

We hope that these revisions adequately response your concerns and enhance the overall quality and impact of our manuscript. We are grateful for the opportunity to improve our work and thank you for the constructive feedback.

We look forward to the possibility of our manuscript being accepted for publication in PLOS ONE and believe it will make a meaningful contribution to the field.

Happy New Year 2024 in Advance!

Sincerely,

Yuxuan Zhou

School of Public Administration, Southwestern University of Finance and Economics, Chengdu, Sichuan, China

Liulin Campus, Southwestern University of Finance and Economics, 555 Liutai Avenue, Wenjiang District, Chengdu, Sichuan, China

E-mail: 15556932867@163.com (Zhou)

On December 26, 2023

---

## [Editor Report · Decision Letter 1]

3 Jan 2024

County Medical Community, Medical Insurance Package Payment, and Hierarchical Diagnosis and Treatment

PONE-D-23-37259R1

Dear Dr. ZHOU,

We’re pleased to inform you that your manuscript has been judged scientifically suitable for publication and will be formally accepted for publication once it meets all outstanding technical requirements.

Kind regards,

Alfredo Luis Fort, M.D., M.Sc., Ph.D.

Academic Editor

PLOS ONE

Additional Editor Comments (optional):

The authors have made the necessary corrections, clarifications and descriptions of necessary aspects of the manuscript, so it is now ready for publication. There are only minor edits required, which can be found in my attached file (plus what is required for publishing purposes).

---

## [Editor Report · Acceptance letter]

27 Mar 2024

PONE-D-23-37259R1 

PLOS ONE

Dear Dr. Zhou, 

I'm pleased to inform you that your manuscript has been deemed suitable for publication in PLOS ONE. Congratulations! Your manuscript is now being handed over to our production team.

Kind regards, 

on behalf of

Dr. Alfredo Luis Fort 

Academic Editor

PLOS ONE